# Effects of Roughage on the Lipid and Volatile-Organic-Compound Profiles of Donkey Milk

**DOI:** 10.3390/foods12112231

**Published:** 2023-06-01

**Authors:** Wei Ren, Mengqi Sun, Xiaoyuan Shi, Tianqi Wang, Yonghui Wang, Xinrui Wang, Bingjian Huang, Xiyan Kou, Huili Liang, Yinghui Chen, Changfa Wang, Mengmeng Li

**Affiliations:** School of Agricultural Science and Engineering, Liaocheng Research Institute of Donkey High-Efficiency Breeding and Ecological Feeding, Liaocheng University, Liaocheng 252000, China; weiren0214@163.com (W.R.); 17862555726@163.com (M.S.); xiaoyuans2021@163.com (X.S.); qitianwang9797@163.com (T.W.); wyh19920729@163.com (Y.W.); wangxinrui0820@163.com (X.W.); 17853148163@163.com (B.H.); kxy19990113@163.com (X.K.); lhl2201091019@163.com (H.L.); chenyinghui0806@163.com (Y.C.); wangcf1967@163.com (C.W.)

**Keywords:** donkey milk, roughage, lipids, volatile organic compounds, LC–MS, GC–MS

## Abstract

The lipid molecules and volatile organic compounds (VOCs) in milk are heavily influenced by diet. However, little is known about how roughage affects the lipid and VOC contents of donkey milk. Accordingly, in the present study, donkeys were fed corn straw (G1 group), wheat hulls (G2 group), or wheat straw (G3 group), and the lipid and VOC profiles of their milk were determined using LC–MS and GC–MS. Of the 1842 lipids identified in donkey milk, 153 were found to be differential, including glycerolipids, glycerophospholipids, and sphingolipids. The G1 group showed a greater variety and content of triacyclglycerol species than the G2 and G3 groups. Of 45 VOCs, 31 were identified as differential, including nitrogen compounds, esters, and alcohols. These VOCs were significantly increased in the G2 and G3 groups, with the greatest difference being between the G1 and G2 groups. Thus, our study demonstrates that dietary roughage changes the lipid and VOC profiles of donkey milk.

## 1. Introduction

Dairy products are among the most common foods in daily life, accounting for 14% of global agricultural trade [1,2]. With ongoing improvements to quality of life, demand for high-quality and diverse dairy products continues to increase. Donkey milk has been demonstrated to be a nutritional foodstuff for the human diet owing to its content of bioactive chemicals, specifically lipids, that either directly or indirectly modulate the intestinal environment’s immunity, influence glucose and lipid metabolism, and lower fat buildup in skeletal muscle [3,4,5].

Milk lipids are not only an important indication of milk quality, they also play an important role in promoting infant brain development, regulate the cell cycle and calcium absorption, reduce inflammatory factors, and reduce skeletal muscle lipid accumulation [6,7,8,9,10]. Milk typically contains 98% triacylglycerols (TGs), 1% phospholipids (PLs), and 0.3–0.5% cholesterols [9,11]. Donkey milk has a low fat content (0.3–1.8 g/100 g), yet it contains up to 30.7% essential fatty acids (FAs) (e.g., linoleic and linolenic acids), which is 25% higher than cow’s milk [5,12]. Therefore, donkey milk is an emerging choice for people seeking a healthy diet.

Previous LC–MS studies have identified 335 lipids from 13 subclasses in donkey milk, including TGs, phosphatidic acids (PAs), phosphatidylcholines (PCs), and phosphatidylethanolamines (PEs) [13]. In addition, the volatile organic compounds (VOCs) in milk are vital indicators by which consumers evaluate the quality of milk and decide whether to purchase [14]. The basic VOCs in milk have been identified as ketones, aldehydes, methyl acids, short-chain alcohols, lactones, pyrazines, and thiols [15,16], while a GC–MS study found that 2-heptanone, 1,3-bis-(1,1-dimethylethyl)-benzene, nonanal, D-limonene, octanoic acid, and 1-octanol are the characteristic VOCs in fresh donkey milk [17].

The lipids and VOCs in milk are influenced by several factors, including metabolic differences in the animal, feed ingredients and additives, biological enzymes, microorganisms, and production processes [18,19]. Importantly, several studies have demonstrated that diet is the most important factor affecting milk lipid and VOC profiles [18,20,21,22]. The quality of the roughage affects the energy content of the feed and accounts for 50%–80% of a donkey’s diet [23]. Many studies have demonstrated that high-quality roughage (herbage, crop straw) increases an animal’s dry-matter intake and affects FA uptake by the mammary glands, which in turn increases milk production, milk protein content, and milk lipid content, resulting in an altered milk lipid profile [24,25]. Roughage’s influence on microbial fermentation and hydrogenation modifies the composition of milk. Specifically, animals inhale VOCs from roughage, which then enter the bloodstream and the milk [18,26]. A previous study revealed that milk from cows fed grass–clover silage accumulates more lipid hydroperoxides and hexanal than milk from cows fed hay [27]. Higher levels of α-linolenic acid and vaccenic acid as well as higher milk yields have been reported for sheep fed hempseed cake than those fed hemp seed [28]. Another study revealed that sheep fed on New Zealand pasture showed a significant increase in γ-12:2 lactone content [22], while Italian forage significantly increased the content of active aldehydes, esters, and terpenoids in cheese VOCs [29]. These studies have shown that VOCs in milk can be modified through feed modification. However, there are surprisingly few studies on the effects of dietary roughages on milk quality, especially in terms of the milk lipids and VOCs in donkey milk. LC–MS and GC–MS are simple and sensitive techniques with high separation capacity and a wide analytical range that are commonly used for the analysis of milk components [30,31,32].

Therefore, the present study aimed to investigate the effects of different dietary roughages on the lipid and VOC profiles of donkey milk using LC–MS and GC–MS. The results presented will increase our understanding of the effects of dietary roughage on donkey milk quality and provide valuable information for the farming of donkeys.

## 2. Materials and Methods

### 2.1. Animal Management and Sample Collection

In this study, all animal experiments were approved by the Animal Care and Use Committee of Liaocheng University (Shandong, China) (2023022706). Thirty-six healthy pluriparous jennies (aged 5 years, average weight: 286 ± 25 kg, and lactation: 60 ± 15 d) were obtained from a local farm (Shandong Province) and reared in the same semi-intensive system. They were randomly allocated to one of three groups (12 donkeys per group). Three kinds of roughage, corn straw (dry matter (DM): 93.70%, crude protein (CP): 4.28%, crude fat (CF): 1.90%, crude ash (Ash): 7.48%, neutral detergent fiber (NDF): 68.30%, acid detergent fiber (ADF): 36.60%, calcium (Ca): 0.57%, phosphorus (P): 0.18%; G1 group), wheat hulls (DM: 92.58%, CP: 5.08%, CF: 1.27%, Ash: 7.03%, NDF: 55.81%, ADF: 34.98%, Ca: 0.34%, P: 0.24%; G2 group), and wheat straw (DM: 89.92%, CP: 4.82%, CF: 1.02%, Ash: 9.21%, NDF: 62.32%, ADF: 36.28%, Ca: 0.31%, P: 0.20%; G3 group) were used as basic dietary roughage. The formulation, proximate analyses, and major FA levels of the diets are presented in Table 1. The donkeys were fed twice daily (9:00 and 16:00) with free access to free water. The study period was two weeks, and the formal period was four weeks. At the end of the study, nine donkeys were randomly selected from each group, and 100 mL milk was taken from each. All the samples were labeled and immediately transferred to the laboratory at −20 °C and stored at −80 °C prior to lipid and VOC analysis.

### 2.2. Lipid Analysis

The process for lipid extraction was based on previous reports with minor modifications. The lipids were extracted from the milk with pre-cooled chloroform/methanol (2:1 *v/v*) at −20 °C as described previously [33,34]. Then, 200 μL of isopropanol was used to dissolve the concentrate, and it was filtered through a 0.22-μm membrane to obtain the analysis sample. To monitor deviations of the analytical results of the mixtures and avoid instrumentation errors, 20 μL from each sample was taken as quality control (QC) samples. The rest of the samples were used for lipidomic assay.

The UPLC–ESI–MS/MS procedure employed was based on previously published methods with minor modifications [35]. The donkey milk was subjected to a comprehensive untargeted lipidomics analysis using a Shim-pack UFLC SHIMADZU CBM30A UPLC system equipped with an Applied Biosystems 6500 Q TRAP mass spectrometer [36]. Chromatographic separation was performed using an ACQUITY UPLC® BEH C18 (100 × 2.1 mm, 1.7 μm, Waters) at a flow rate of 0.25 mL/min and a column temperature of 50 °C. The mobile phases were 60% acetonitrile + 40% water (10 mM ammonium formate + 0.1% formic acid) (A) and 90% isopropanol + 10% acetonitrile (0.1% formic acid +10 mM ammonium formate) (B). The elution gradient was: 0–5 min, 70–57% A; 5–5.1 min, 57–50% A; 5.1–14 min, 50–30% A; 14–14.1 min, 30% A; 14.1–21 min, 30–1% A; 21–24 min, 1% A; 24–24.1 min, 1–70% A; 24.1–28 min, 70% A. An electrospray ionization source (ESI) was used in both positive and negative ionization modes with a positive ion spray voltage of 3.50 kV and a negative ion spray voltage of 2.50 kV. The capillary temperature was 325 °C. Full scanning was performed at a mass resolution of 35,000 over an m/z range of 150–2,000. Data dependent acquisition (DDA) MS/MS experiments were carried out using higher-energy collisional dissociation for secondary cleavage, with the normalized collision energy of 30 eV. Dynamic exclusion was used to remove unnecessary information from the MS/MS spectra.

### 2.3. HS–SPME–GC–MS Analysis

A donkey milk sample (15 mL) was placed in a 20 mL headspace vial (Agilent, Palo Alto, CA, USA). An SPME fiber (120 µm DVB/CWR/PDMS) was exposed to the headspace for 15 min at 100 °C. Then, they were subjected to GC (Agilent, 8890GC-5977B, Santa Clara, CA, USA) desorption of the VOCs at 250 °C for 5 min in splitless mode. A 30 m × 0.25 mm × 0.25 μm DB-5MS (Agilent J&W Scientific, Folsom, CA, USA) capillary column was used for separation. The carrier gas utilized had a linear velocity of 1.2 mL/min, and the injector temperature was 250 °C. The column oven temperature was kept at 40 °C (3.5 min), then increased at 10 °C/min to 100 °C, 7 °C/min to 180 °C, and 25 °C/min to 280 °C, where the temperature was held for 5 min. Mass spectra were recorded in electron impact ionization mode at 70 eV and scanned at 1-s intervals in the *m*/*z* 50–500 amu range. The temperatures of the quadrupole mass detector, ion source, and transfer line were set to 150, 230, and 280 °C, respectively. The NIST mass spectrometry library was used to identify the VOCs (retention indices) and quantitatively analyze the samples.

### 2.4. Statistical Analysis

The data were processed using SPSS version 26.0 (SPSS Inc., Chicago, IL, USA), subjected to one-way analysis of variance (ANOVA), and compared for differences by Tukey’s test, Duncan’s multiple range test, and Fisher’s protected least significant difference (LSD) test. The lipidomic data were expressed as mean ± standard error of the mean (SEM, *n* = 9) with a significant difference in variable importance in projection (VIP) > 1 and *p* < 0.05. The data were assessed by orthogonal partial least squares discriminant analysis (OPLS-DA). Metabolites and pathways associated with the differential lipid compounds were identified using the Kyoto Encyclopedia of Genes and Genomes (KEGG) pathway database (https://www.kegg.jp/kegg/, accessed on 20 April 2023). An amount of *p* < 0.05 was taken to indicate significant differences in the VOCs.

## 3. Results

### 3.1. Lipid Profiles of Donkey Milk

Twenty-seven samples and eight QC samples were analyzed under positive and negative ion mode (Appendix A). The principal component analysis score chart of donkey milk lipid QC shows that QC samples are clustered, with good repeatability and stability, indicating the reliability of the data (Appendix A). A total of 1841 lipids from 29 subclasses were detected in the milk samples (Figure 1A). The lipids were assigned to six categories [64.86% glycerolipids (GLs), 17.71% glycerophospholipids (GPs), 8.47% sphingolipids (SPs), 2.18% sterol lipids (ST), 1.14% FAs, and 5.64% derivatized lipids (DEs)]. In terms of subclass, at 55.19%, TG was discovered to be the dominant lipid, followed by 8.69% PC, 8.20% diglyceride (DG), and 5.98% PE (Figure 1B). The relative amounts of TG and monoglyceride (MG) in the G1 group were significantly higher than those in the G3 group, while the contents of PC, PE, phosphatidylglycerol (PG), phosphatidylinositol (PI), ceramide (Cer), sphingomyelin (SM), zymosterol (ZyE), and bis-methyl phosphatidic acid (BisMePA) in G3 were significantly higher than those in the G1 group (*p* < 0.05, Table 2).

### 3.2. Differential Analysis of Lipids

The OPLS-DA scatter plots reveal distinct differences in milk lipids between the G1, G2, and G3 groups (Figure 2A). The R2 and Q2 intercept values are (0.0, 0.62) and (0.0, −0.77), which are robust and not over-fitted (Figure 2B). Specifically, 153 lipids were identified as significantly different between the G1, G2, and G3 groups, including 126 TGs, 6 DGs, 6 Cers, and 9 PLs by applying VIP > 1 and *p* < 0.05 (Appendix A and Figure 2C). In addition, extensive associations between DG, BisMePA, and Cer and other lipid species, especially TG, were observed (Figure 2D).

### 3.3. Lipid Metabolism Pathways

Based on KEGG analysis, 17 metabolic pathways were identified for the 153 differential lipids, of which the GL metabolism pathway was dominant, with the greatest number of significantly different lipids (Figure 3A). Seven metabolic pathways were found to be most relevant for the lipids, including GL metabolism, SP metabolism, linoleic acid metabolism, GP metabolism, α-linolenic acid metabolism, arachidonic acid metabolism, and GPI-anchor biosynthesis (Figure 3B).

### 3.4. Qualitative and Quantitative Analysis of VOCs

A total of 45 VOCs were identified in the donkey milk, belonging to 10 categories: 15 hydrocarbons, 6 heterocyclics, 6 ketones, 5 nitrogen compounds, 3 phenols, 2 aldehydes, 2 sulfur compounds, 1 alcohol, 1 ester, and 4 others (Figure 4A). The heterocycles dominated the VOCs, followed by nitrogen compounds and esters (Figure 4B). Nitrogen compounds, phenols, ketones, and hydrocarbons were significantly higher in the G1 group than the G2 and G3 groups, while the opposite was true for esters and alcohols (*p* < 0.05, Figure 4C). In addition, the most prevalent substances in the G1 group was 4-amino-2(1h)-pyridinone (298.40 ± 7.82 ng/g), while 4,6-dimethyl-isothiazolo [5,4-b]pyridin-3-one (341.38 ± 19.97 ng/g and 379.18 ± 40.94 ng/g, respectively) was the substance with the highest concentration in the G2 and G3 groups (Appendix A).

### 3.5. Differential VOCs

The OPLS-DA scatter plots show a distinct separation trend and clustering (Figure 5A). The R2 (0.0, 0.6) and Q2 (0.0, −0.89) scores show that the model’s fitness was strong and not over-fitted (Figure 5B). A total of 31 significantly different VOCs were identified in the samples based on *p* < 0.05 (Figure 5C). 1-Methyl-1-hydroxymethyladamantane, 1-ethyl-2,4-dimethyl-benzene, 3-(acetyloxymethyl)-2,2,4-trimethyl-cyclohexanol, methyl ester-3-nonenoic acid, 4-(1,1-dimethylethyl)-benzenepropanal, 1-(1-cyclohexen-1-yl)-1-propanone, 4,4,5-trimethyl-2-cyclohexen-1-one, 5-methyl-5-(1-methylethyl)-3-heptyne-2,6-dione, 4-amino-2(1h)-pyridinone, methyl 5-hydroxymethyl-4-imidazolecarboxylate, 4-methyl mercaptoaniline, dimethyl-sulfur diimide, octahydro-3-methyl-1h-indole, 1-(2-furanyl)-1-propanone, N-(1,1-dimethylethyl)-2-benzoxazolamine, 4,6-dimethyl- isothiazolo [5,4-b]pyridin-3-one, 6-methyl-7-oxa-8-azabicyclo [4.2.1]non-8-ene, 1-iodo-dodecane, bromo-methane, and iodo-methane levels were significantly higher in the G2 and G3 groups than in the G1 group, while 2-methyl-nonane and ethyl ester hydrazinecarboxylic acid were significantly lower (*p* < 0.05, Appendix A).

## 4. Discussion

Donkey milk has a total protein content of 1.5–1.8%, a lactose content of 5.8–7.4%, and a fat content of 0.2–1.8% [37,38]. Lipids are one of the main nutrients in donkey milk and an important indicator of milk quality in commerce [39,40]. Mice fed donkey milk, especially the n-3 family, were found to show increased hepatic mitochondrial uncoupling, decreased energy efficiency, maintained energy balance, decreased body fat deposition, and improved antioxidant and anti-inflammatory defenses [41]. Therefore, donkey milk lipids represent a valid focus of research. This study employed UPLC–ESI–MS/MS, which is considered a novel cutting-edge technology that provides a platform for evaluating the lipids in donkey milk primarily on a minor scale with excellent precision and in a short period of time [42]. In this study, 1842 lipids were detected in the donkey milk. This is significantly higher than that for donkey milk (335 lipids), cow milk (557 lipids), goat milk (756 lipids), and human milk (617 lipids) identified by conventional LC–MS [13,43,44,45]. Specifically, MG, PG, BisMePA, wax ester (WE), ZyE, Co, and AcHex were newly identified in the present study compared with previous studies [13,37,46,47]. Clearly, the lipids detected in this study are more abundant than those previously studied, which could be due to the technological superiority of the technique used. TGs (a total of 1016) were discovered to be the dominant lipids in donkey milk, which is consistent with the results of previous studies [13,48]. Interestingly, we detected PA and SiT in the dietary roughage, but they were not detected in the donkey milk. This may be due to the degradation of PA to DG [49] by phospholipid hydrolase.

Previous studies have shown that different fermentation systems present different FA compositions, and the production rate of milk lipids increases with FA concentration [50,51]. Dietary fiber has an impact on how the lipids are metabolized, which results in further differences in the lipid compositions of milk [52,53]. A previous study indicated that the contents of specific FAs in TG molecules is positively correlated with the content of the corresponding dietary FAs [54]. The levels of PC, PE, PG, Cer, hexosylceramide, SM, and BisMePA in the G1 and G2 groups were significantly lower than those in the G3 group, suggesting that these lipids are affected by roughage. Interestingly, TGs, especially TG (18:4-8:0-18:1) and TG (8:0-10:0-18:3), were observed at higher levels in the G1 group than in the G2 and G3 groups. It is possible that the roughage in the G1 group contained more PUFAs, enhancing the diversity and content of TG species [55]. In the present study, the G2 and G3 groups contained higher levels of C16:0 in their diets, which resulted in higher levels of C16:0 in the milk lipids. This is consistent with the finding of a previous study, i.e., that changes in the equivalent FA content of milk fat can come from changes in the lipid composition of the crude feed [56]. In addition, monogalactosylmonoacylglycerol (MGMG), monogalactosyldiacylglycerol (MGDG), and digalactosyldiacylglycerol (DGDG) were detected in the donkey milk, and these lipids are present in wheat flour [57]. The main lipid class contents of the different groups have linear relationships with their dietary lipid profiles, and this phenomenon has been observed upon the modification of lipid intake in human, sheep, and cow diets [51,53,58,59]. TGs were found to be most interactive with other lipid species, which indicates that TG may play an essential role in milk production in donkeys fed on different roughages. GL metabolism, sphingolipid metabolism, linoleic acid metabolism, GP metabolism, α-linolenic acid metabolism, arachidonic acid metabolism, and GPI-anchor biosynthesis have the most impact on the 153 significant different lipids. These results are consistent with those of a previous study on the lipid metabolism pathways in donkey colostrum and mature milk [13]. Overall, the lipid compositions of the roughage have a great influence on the lipids in donkey milk.

VOCs are the primary driver of food purchases by consumers. The HS–SPME–GC–MS method is widely used for qualitative and quantitative analysis of VOCs in dairy products [60]. In this study, 45 VOCs (15 hydrocarbons, 6 heterocyclics, 6 ketones, 5 nitrogen compounds, 3 phenols, 2 aldehydes, 2 sulfur compounds, 1 alcohol, 1 ester, and 4 others) were detected in the donkey milk. These results are similar to those from studies on nut-based milk, skimmed milk, and donkey milk powder, which were reported to contain 13 VOCs (ketones, acids, aldehydes, alcohols, alkanes, and sulfur compounds), 70 VOCs (aldehydes, ketones, and alcohols), and 39 VOCs (48.89% acids, 0.53% alcohols, 28.26% aromatic compounds, 9.22% ketones, 5.04% aldehydes, 5.39% alkanes, 1.39% esters, and 1.39% others), respectively [61,62,63,64]. In addition, nitrogen compounds, esters, alcohols, and phenols were the most abundant VOCs, and there are significant differences among the three groups, suggesting that these VOCs were affected by dietary roughage [65,66].

A previous study revealed that VOC levels are increased in milk from sheep fed free-range hay, while VOCs remain unchanged by added concentrate content (commercial concentrate mixture of Formel 97 L˚ag at 0.2 kg/day and 0.7 kg/day, respectively) [67]. Free fatty acids and ketones were increased in milk from animals fed on ensiled mycorrhizal forage [68]. Higher levels of camphene, sabinene, β-caryophyllene, and skatole have been observed in the milk of cattle fed a pasture-derived diet than those fed a hay-based diet [69]. In the present study, 21 VOCs were significantly decreased in the milk of donkeys fed corn straw, while the 2-methyl-nonane and ethyl ester hydrazine carboxylic acid contents were increased. 3,3-Dimethyl-undecane, nonanal, and 2-heptanone were increased in the milk of donkeys fed wheat hull. In donkeys, chemicals in the feed can enter the mammary tissue directly from the circulation and undergo biochemical reactions in the blood to form new biologically active substances, or they can be transferred directly through the blood to the mammary tissue to produce milk [18,70,71]. Generally, milk from pasture-fed animals has more odor-active molecules than milk from animals fed total mixed ration [18,70]. These results provide valuable insight into the quality of donkey milk and a new perspective on feeding lactating donkeys.

## 5. Conclusions

There were differences between milk lipids and volatile organic compounds as a result of changes in roughage. A total of 1842 lipids were detected, among which 153 lipids were considered differential, being mainly involved in GP and SP metabolism pathways. There was significantly increased GL content in the milk of donkeys fed corn straw, while GP and SP contents were significantly increased by feeding with wheat straw. The result indicates that coarse grains mainly influence the GL, GP, and SP composition of donkey milk lipids. Furthermore, 31 out of 45 VOCs were determined to be differential, mainly esters, alcohols, and phenols, the contents of which were significantly increased for donkeys fed wheat hull and wheat straw than corn straw.

## Figures and Tables

**Figure 1 foods-12-02231-f001:**
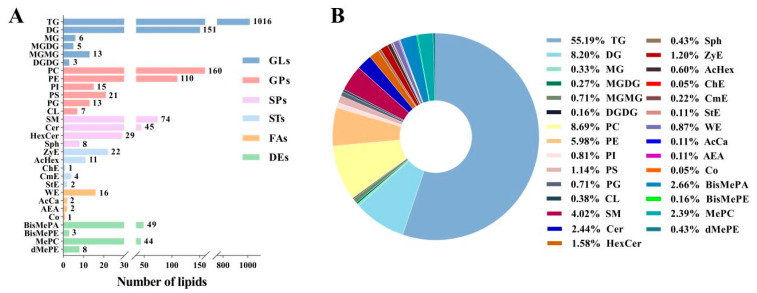
Qualitative analysis results of lipids in donkey milk. Numbers (**A**) and percentages (**B**) of lipid subclasses.

**Figure 2 foods-12-02231-f002:**
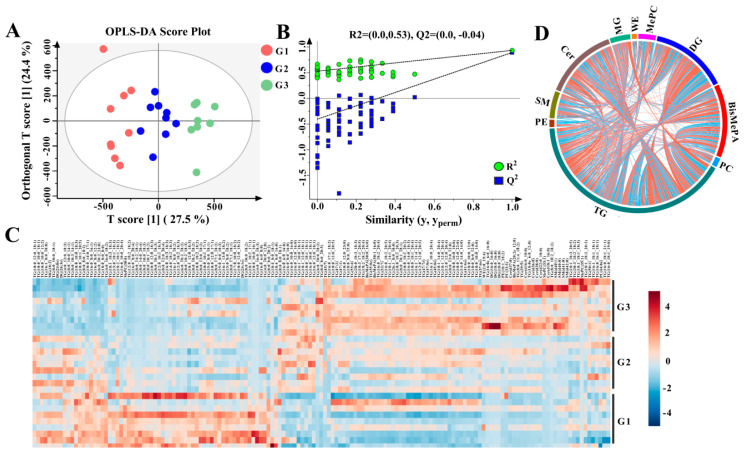
Difference analysis results for lipids between the G1, G2, and G3 groups. OPLS-DA plot (**A**); corresponding OPLS-DA validation plots (**B**); heatmap of hierarchical clustering analysis (red indicates increased, blue indicates decreased) (**C**); chord diagram of differential lipid associations (positive correlation in red, negative correlation in blue) (**D**).

**Figure 3 foods-12-02231-f003:**
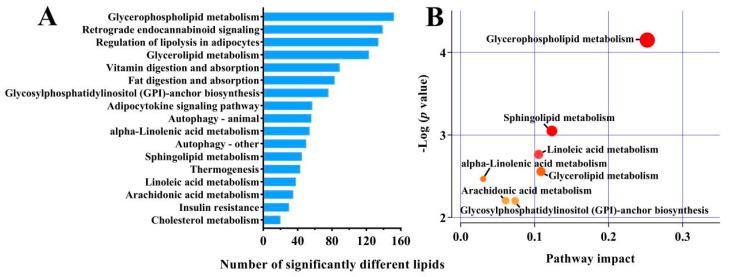
KEGG pathways significantly enriched in different lipid species for the G1, G2, and G3 groups (**A**); metabolomic map of the important metabolic pathways involving differential lipids between G1, G2, and G3 (**B**), sizes indicate the major pathway enrichments; colors indicate high pathway impact values.

**Figure 4 foods-12-02231-f004:**
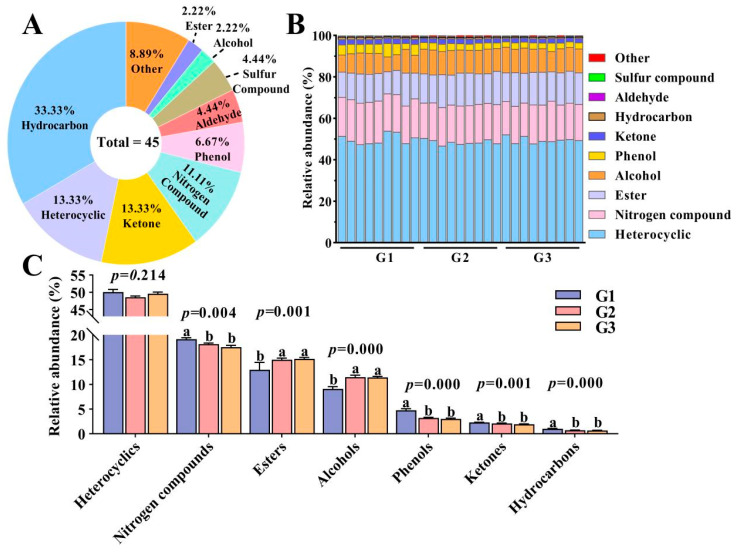
Qualitative analysis results for VOC compositions of different donkey milks. Quantity percentages of VOCs for G1, G2, and G3 (**A**); content percentages of VOCs (the height of the rectangle indicates the percentage of each subclass in relation to the current item) (**B**); relative VOC contents (% total VOC composition) for G1, G2, and G3 (**C**), values shown as mean ± SME (*n* = 9), where different letters indicate significant difference (*p* < 0.05).

**Figure 5 foods-12-02231-f005:**
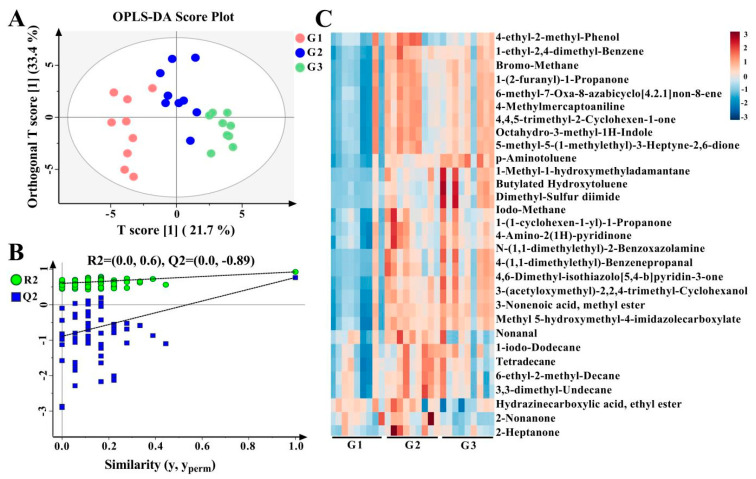
Difference analysis of VOC compositions between G1, G2, and G3. OPLS-DA plot (**A**), corresponding OPLS-DA validation plots (**B**), heatmap of hierarchical clustering analysis (red indicates increased, blue indicates decreased) (**C**).

**Table 1 foods-12-02231-t001:** Ingredients, proximate analyses, and fatty acid profiles of experimental diets.

Item	G1	G2	G3
Ingredients (%)			
Corn	15.50	15.50	15.50
Soybean	3.10	3.10	3.10
Salt	0.20	0.20	0.20
Premix ^1^	1.20	1.20	1.20
Corn straw	80.00	0.00	0.00
Wheat hulls	0.00	80.00	0.00
Wheat straw	0.00	0.00	80.00
Total	100.00	100.00	100.00
Proximate composition (% dry matter)			
Dry matter	95.49	93.53	91.48
Crude protein	6.30	6.50	6.12
Crude fat	2.66	2.33	2.31
Crude ash	11.50	11.14	12.06
Major fatty acids (% total fatty acid)			
C16:0	18.26	20.96	19.90
C18:0	3.27	3.47	5.39
C18:1	24.03	23.95	26.68
C18:2n-6	49.16	46.60	43.43
C18:3n-6	4.46	4.18	4.28

Ingredients, proximate composition, and major fatty acids were measured and analyzed based on triplicate determination. ^1^ Premix/kg: vitamin A; 275 KIU; vitamin D3; 115 KIU; vitamin E > 180 IU; Fe (green vitriol); 3500 mg; Cu (copper sulfate); 725 mg; Zn (zinc sulfate); 2500 mg; Mn (manganese sulfate); 1750 mg; I (calcium iodide); 54 mg; Se (sodium selenite); 7.3 mg; Methionine ≥ 0.2%; Lysine ≥ 1%.

**Table 2 foods-12-02231-t002:** Lipid compositions of experimental diets and donkey milk.

Lipid	Diet (%)			Donkey Milk (%)			SEM	*p* Value
	G1	G2	G3	G1	G2	G3		
TG	68.10	68.21	66.17	96.06 ^a^	95.86 ^a^	94.34 ^b^	0.23	0.001
DG	3.13	3.58	3.27	1.83	2.02	1.83	0.05	0.184
MG	0.0445	0.0346	0.0366	0.0935 ^a^	0.0820 ^ab^	0.0631 ^b^	0.0052	0.047
PC	10.48	9.49	14.16	0.40 ^b^	0.38 ^b^	0.77 ^a^	0.05	0.001
PE	0.47	0.36	0.44	0.20 ^b^	0.19 ^b^	0.41 ^a^	0.03	0.003
PS	0.3331	0.3245	0.2632	0.0022	0.0012	0.0024	0.0004	0.409
PG	0.31	0.4849	0.2659	0.0072 ^b^	0.0088 ^b^	0.0163 ^a^	0.0012	0.001
PI	0.1685	0.1956	0.1232	0.0010 ^b^	0.0016 ^ab^	0.0019 ^a^	0.0001	0.013
PA	0.0262	0.0389	0.0234	-	-	-	-	-
Cer	6.37	5.20	3.34	0.14 ^b^	0.26b	0.47 ^a^	0.04	0.002
HexCer	3.518	1.9621	2.1496	0.0325	0.0255	0.0431	0.0033	0.079
CL	0.0324	0.0297	0.0195	0.0012	0.0004	0.0011	0.0003	0.49
SM	0.0368	0.0547	0.0761	0.3661 ^b^	0.3073 ^b^	0.7299 ^a^	0.06	0.001
Sph	0.0468	0.0466	0.0361	0.0079	0.0172	0.027	0.0036	0.094
Co	0.1827	0.1125	0.152	0.0008 ^ab^	0.0006 ^b^	0.0011 ^a^	0.0001	0.032
WE	0.1539	0.6102	0.5052	0.0754	0.1304	0.2161	0.0248	0.059
AcHex	4.8445	6.1485	7.2537	0.0327 ^a^	0.0230 ^b^	0.0339 ^a^	0.0019	0.027
ZyE	0.17	0.49	0.17	0.12 ^b^	0.15 ^a^	0.15 ^a^	0.00	0.003
StE	0.0346	0.1441	0.0383	0.0429	0.0376	0.0327	0.0028	0.534
SiE	0.0556	0.255	0.0776	-	-	-	-	-
MePC	0.14	0.10	0.18	0.13	0.05	0.25	0.04	0.085
BisMePA	0.5683	0.6632	0.5937	0.0004 ^b^	0.0003 ^b^	0.0007 ^a^	0.0228	0.018

TG, triglyceride; DG, diglyceride; MG, monoglyceride; CL, cardiolipin; PA, phosphatidic acid; LPC, lysophosphatidylcholine; PC, phosphatidylcholine; PE, phosphatidylethanolamine; PG, phosphatidylglycerol; PI, phosphatidylinositol; PS, phosphatidylserine; Cer, ceramides; HexCer, hexosylceramide; SM, sphingomyelin; SiE, sitosterol ester; StE, sterol ester; ZyE, zymosterol ester; Co, coenzyme; WE, wax esters; AcHex, diacylatehexosyl; MePC, methyl phosphatidylcholine; Sph, Sphingolipid; BisMePA, bis-methyl phosphatidic acid; “-” indicates not detected. ^a,b^ Means within a row with different superscripts differ (*p* < 0.05).

## Data Availability

The data presented in this study are available on request from the corresponding author.

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
