# Peer review of "Effects of Roughage on the Lipid and Volatile-Organic-Compound Profiles of Donkey Milk"

_foods, 2023, doi:10.3390/foods12112231_

Round 1

Reviewer 1 Report

Dear Editor and Authors,

I send you my review about the article entitled “Effects of roughage on the lipid and volatile-organic-compound profiles of donkey milk”.

The aim of the paper, as reported in the scope was to investigate the effects of different dietary roughages on the lipid and volatile organic compounds profiles of donkey milk.

In my opinion, although the Article is well written, in a good English language and it is well structured, it show, also, some lacks that I reported below.

In the introduction should be better explained the originality of this Article. To improve this aspect, I suggest to the Authors, to add some comments about the articles cited from line 64 to line 65.

Moreover the sentence from line 70 to line 71 should be shift in the conclusions.

The paragraph materials and methods result complete, however, in the paragraph “2.4. statistical analysis”, it should reported the model used in the ANOVA to obtain the least square means.

Moreover, It should be explained why if the authors have allocated 12 jennies in each group they considered only 9 jennies for the collection of milk samples.

The results is well presented and they are well discussed, also in comparison to the data reported in the literature.

Finally, the conclusions appear to be too synthetic respect the data showed and to the aim of the research, in my opinion, it should best stress the relevance of the findings of this research.

Best regards

Author Response

Response to Reviewer1 comments

We are grateful for the comments from both the editor and reviewer1. They were very helpful and have enabled us to revise and improve our manuscript (Manuscript ID: 2416170). The point-by-point response1 to each comment are provided below. Additionally, some other revisions have been made, and are highlighted in the revised manuscript.

Reviewer 1

In my opinion, although the Article is well written, in a good English language and it is well structured, it show, also, some lacks that I reported below.

In the introduction should be better explained the originality of this Article. To improve this aspect, I suggest to the Authors, to add some comments about the articles cited from line 64 to line 65.

Re: Thank you for the comments. The comments about the articles cited from line 64 to line 65 have added in the revised manuscript (Lines 65-66).

The sentence from line 70 to line 71 should be shift in the conclusions.

Re: The sentences from line 70 to 71 have been shifted.

The paragraph materials and methods result complete, however, in the paragraph “2.4. statistical analysis”, it should reported the model used in the ANOVA to obtain the least square means.

Re: In the paragraph “2.4. statistical analysis”, the model used in the ANOVA to obtain the least square means has been described (Lines 177-179).

It should be explained why if the authors have allocated 12 jennies in each group they considered only 9 jennies for the collection of milk samples.

Re: Referring to previous experiments, in large animal experiments such as donkeys, we usually believe that six or more experimental animals can ensure the scientific nature of the experimental results. Therefore, data from nine jennies (9 jennies were randomly selected from 12 jennies) can be used for experimental analysis.

Finally, the conclusions appear to be too synthetic respect the data showed and to the aim of the research, in my opinion, it should best stress the relevance of the findings of this research.

Re: The conclusions has been added the relevance of the results of this study.

Reviewer 2 Report

It is scientifically useful research. It will contribute to science. However, the 1842 lipid identification is much higher than previous studies. This result should be explained in more detail, except that it is tied to the method. Or I think it requires a repetition. Preliminary essay can be written in the title.

Also line 83-85: why 9 samples were chosen instead of 12 is not clear. If the samples are in the deep freezer, 12 milks should be examined in every 3 feeds.

Author Response

Response to Reviewer2 comments

We are grateful for the comments from both the editor and reviewer 2. They were very helpful and have enabled us to revise and improve our manuscript (Manuscript ID: 2416170). The point-by-point responses to each comment are provided below. Additionally, some other revisions have been made, and are highlighted in the revised manuscript.

Reviewer 2

It is scientifically useful research. It will contribute to science. However, the 1842 lipid identification is much higher than previous studies. This result should be explained in more detail, except that it is tied to the method. Or I think it requires a repetition. Preliminary essay can be written in the title.

Re: The technology has been widely applied in other fields (meat, oil, and nuts), and it is a valuable tool for detecting analytes present at trace levels and screening large numbers of samples. And this technique has been proven to have higher sensitivity, higher resolution and a shorter run-time (https://doi.org/10.1111/1541-4337.12603; https://doi.org/10.1016/j.lwt.2023.114426). In addition, we take 20 μL from each sample for the quality control (QC) samples. These QC samples were used to monitor deviations of the analytical results from these pool mixtures and compare them to the errors caused by the analytical instrument itself.

Also line 83-85: why 9 samples were chosen instead of 12 is not clear. If the samples are in the deep freezer, 12 milks should be examined in every 3 feeds.

Re: In large animal experiments such as donkeys, we usually believe that six or more experimental animals can ensure the scientific nature of the experimental results. Therefore, data from nine jennies (9 jennies were randomly selected from 12 jennies) can be used for experimental analysis.

Reviewer 3 Report

After careful study of your manuscript entitled "Effects of roughage on the lipid and volatile-organic-compound profiles of donkey milk",  I have the following comments to highlight.

I think it would be useful for the authors to include one or more
typical UPLC–ESI–MS/MS chromatograms from lipids fractions of the samples, such as (GLs), glycerophospholipids (GPs), sphingolipids (SPs), sterol lipids (ST), FAs, and derivatized lipids (DEs).
We consider it necessary as there is no similar chromatogram even in the previous similar works mentioned (33,34,35).
The chromatograms from lipid fractions apart from confirming the results, will provide useful information to the readers of the journal.
Also, for a clearer description and distinction between the three trials of experimental diets, we consider the chemical composition of Maize straw, Wheat shell, and Wheat straw necessary.
An additional note:
In the SPME determination method I have to note that the amount of 10 ml sample in the 20 ml sampling vial makes it difficult and problematic to absorb the volatile components by SPME fiber from the sample and further analyze the volatile components.
Could the authors provide a convincing explanation of how this was achieved?

Author Response

Response to Reviewer 3 comments

We are grateful for the comments from both the editor and reviewer 3. They were very helpful and have enabled us to revise and improve our manuscript (Manuscript ID: 2416170). The point-by-point responses to each comment are provided below. Additionally, some other revisions have been made, and are highlighted in the revised manuscript.

Reviewer 3

I think it would be useful for the authors to include one or more typical UPLC–ESI–MS/MS chromatograms from lipids fractions of the samples, such as (GLs), glycerophospholipids (GPs), sphingolipids (SPs), sterol lipids (ST), FAs, and derivatized lipids (DEs).

We consider it necessary as there is no similar chromatogram even in the previous similar works mentioned (33,34,35).

Re: Thank you for the comments. The UPLC–ESI–MS/MS chromatograms has been added supplementary material (Figure S1).

The chromatograms from lipid fractions apart from confirming the results, will provide useful information to the readers of the journal.

Re: In MS based lipidomics research, quality control (QC) is usually carried out in order to obtain reliable and high-quality lipidomics data. The Principal component analysis score chart of donkey milk lipid QC sample has been added supplementary material (Figure S2).

Also, for a clearer description and distinction between the three trials of experimental diets, we consider the chemical composition of Maize straw, Wheat shell, and Wheat straw necessary.

Re: Maize straw [dry matter (DM): 93.70%, crude protein (CP): 4.28%, crude fat (CF): 1.90%, crude ash (Ash): 7.48%, neutral detergent fiber (NDF): 68.30%, acid detergent fiber (ADF): 36.60%, calcium (Ca): 0.57%, phosphorus (P): 0.18%], wheat shell (DM: 92.58%, CP: 5.08%, CF: 1.27%, Ash: 7.03%, NDF: 55.81%, ADF: 34.98%, Ca: 0.34%, P: 0.24%), and wheat straw (DM: 89.92%, CP: 4.82%, CF: 1.02%, Ash: 9.21%, NDF: 62.32%, ADF: 36.28%, Ca: 0.31%, P: 0.20%) has been added lines 82-88.

In the SPME determination method I have to note that the amount of 10 ml sample in the 20 ml sampling vial makes it difficult and problematic to absorb the volatile components by SPME fiber from the sample and further analyze the volatile components. Could the authors provide a convincing explanation of how this was achieved?.

Re: In response to your question, we have rechecked and found that the correct approach is to place a 15 ml sample of donkey milk in a 20 ml headspace vial.
